# Figs (*Ficus carica* L.) Used as Raw Material for Obtaining Alcoholic Fermented Beverages

**Emilia Moisescu and Arina Oana Antoce *** 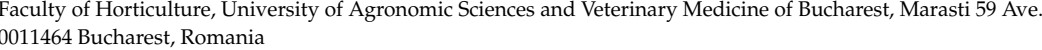

Faculty of Horticulture, University of Agronomic Sciences and Veterinary Medicine of Bucharest, Marasti 59 Ave., 0011464 Bucharest, Romania
* Correspondence: aantoce@yahoo.com

**Abstract:** The fig tree is one of the oldest species cultivated by mankind. In spite of having high nutraceutical value and a pleasant taste, the fig is not a widely cultivated fruit. Furthermore, figs are extremely perishable, therefore they are usually sold as dried fruits rather than fresh. To harness this valuable fruit, other derived products should also be considered. For instance, the production of alcoholic beverages fermented from figs comes in response to the interest of producers to capitalize on these fruits in other ways than as fresh or dried fruit or jam. The present research focuses on the possibility of obtaining marketable fermented beverages made from three fig varieties cultivated in southwestern Romania. The aim of the study was to provide an optimized technological process for the production of alcoholic beverages fermented from fresh figs and to assess their quality and acceptability. The products were obtained in triplicate from each fig variety and their quality was assessed by determining their main chemical parameters, as well as their sensory properties. This research provided valuable information regarding the technological process to be applied for fig fermented beverages, processes which can still be further refined to better meet the consumer demand.

**Keywords:** chemical parameters; fig variety; technological process; sensory properties





## 1. Introduction

Figs, the fruits of *Ficus carica* from the *Moraceae* family, are among the oldest fruit species domesticated and are cultivated worldwide, Turkey being the main producer [1]. The fig tree grows easily in warm and temperate climates, is tolerant to drought, does not require rich soils and is resistant to pests.

Figs are eaten fresh when in season and are sought by consumers for their fine taste. Fresh figs have a honey-like sweetness with a mild acidity and some fruity aroma remi­niscent of berries and grapes. However, because they are very perishable fruits, especially due to postharvest fungal growth [2,3], figs are mostly exploited dehydrated or processed into jams or other sweet products. Some packaging to preserve the fresh fruits and their qualities for longer have also been attempted [4], but the practice has not spread.

Even though figs have some composition and aroma similarities to grapes (the fruits of *Vitis vinifera*) and in spite of their presence in regions where the grapevine is also cultivated, they were not used too much for production of fermented alcoholic beverages, unlike grapes, which led to the well-known product, wine. Some ancient so-called fig wine recipes were preserved and reported [5], but they are not using a typical technology and those products were added with other herbs, salt and so on, most likely for preservation. Nowadays, some recipes are provided on some websites for home-brewing and even some commercial artisanal products are present on the market from time to time, but, to our knowledge, there is no established technology for a fermented beverage with a reliable constant quality.

Nevertheless, being a fruit with a sufficiently high amount of carbohydrate com­pounds, attempts were made to use the figs for the production of fermented beverages

such as vinegar [6,7], lactic fermented beverages [8,9], some alcoholic beverages [10,11], as well as distillates [12].

Fig fresh fruits contain about 79% *w/w* water, 16% *w/w* fermentable carbohydrates (sugars), 3% *w/w* non-fermentable carbohydrates (dietary fibres), 1% *w/w* protein and, as compared to other fruits, only low amounts of acids (0.1–0.3% *w/w*), polyphenols (up to 1.70 mg gallic acid equivalents $g^{-1}$) and other components [13,14] such as vitamin C (from 0.8 to 9.0 mg 100 $g^{-1}$) [14], being comparable to grapes, thus being a potentially good raw material for the obtainment of wine-like fermented alcoholic beverages. However, the carbohydrates from figs are not entirely fermentable, part of them being polymers which do not hydrolyse. Therefore, the beverage which can be obtained from the fermentation of figs' natural carbohydrates without any correction is not sufficiently alcoholic to ensure microbiological stability and aroma conservation, as it is in the case of wine.

A lot of *Ficus* species are reported to have medicinal or dietary usages, among which figs are the most used [15]. The positive effects of figs on human health [16] are based on their antioxidant capacity [17] and other nutraceutical qualities [18,19], mainly due to the presence of polyphenolic compounds such as flavone, rutin and quercetin [20], as well as anthocyanins in the case of coloured fruits. These polyphenolic compounds are extracted also in the beverages produced from the fruits, which is also the case in other fermented beverages [21], higher concentrations being determined for those from blueberries, blackberries and black mulberries (above 1 g $L^{-1}$ GAE-gallic acid equivalents) and lower for quince, apples, apricots, melons, raspberries, cherries and strawberries.

Fig polyphenolic content is low to intermediate, higher concentrations being present in purple cultivars, which contain anthocyanins as colour pigments [1], with cyanidin-3-O-rutinoside being the main anthocyanin [22,23].

Fig nutraceutical value and antioxidant properties are determined mostly by its polyphenolic content [24], but these compounds, especially the colour pigments, are partially lost during fig drying—the most common and facile preservation technique [1]. Conversely, fermenting figs is a better alternative for preserving their naturally accumulated polyphenols, because they are intensely extracted from the skins, especially in the presence of the forming alcohol during the process of maceration–fermentation.

The antioxidant capacity of fermented alcoholic beverages in figs is also influenced by the types of yeast used in the fermentation process. Thus, the use of non-*Saccharomyces* yeasts, such as *Pichia fermentans, Wickeromomyces anomalus, Hanseniaspora uvarum*, in co-fermentation processes, determined a higher anthocyanin content and antioxidant capacity than in the case of beverages obtained by using *Saccharomyces cerevisiae* yeasts, even though *Saccharomyces* have a higher fermentation yield and produce beverages with a higher alcohol concentration by 2–3% *v/v* vol. [25].

The fermentation yeasts also very much influence the aroma of the fermented beverage. The flavour compounds identified in the fermented alcoholic beverages from figs obtained by co-fermentation using non-*Saccharomyces* yeasts were higher in number, compared to those produced in the presence of *S. cerevisiae*, 38 vs. 30 [25]. Significant composition differences were found in distillates from fig-fermented beverages obtained with natural versus selected yeasts. Lower methanol and esters were produced by selected yeasts, ensuring a high quality of the final product, with higher sensory analysis scores for taste and odour [12]. Similarly, studies on grape wine showed as well that the use of selected yeast is improving the aromatic profile after fermentation as compared to natural yeasts [26].

The inoculum size and the fermentation period influence the characteristics of the fermented alcoholic beverage in figs. In the study by Pratik in 2010 [27] fermentation variants were made using *Saccharomyces cerevisiae* inoculum of 5, 7.5 and 10% *w/v*, for fermentation periods of 10, 15 and 20 days, at a temperature of 24 °C. The obtained beverages had alcoholic concentrations between 9.9–11.8% *v/v*, 0.44–0.64% *w/v* titratable acidity, 0.4–4.2% *w/v* total sugars, 0.2–4.0% *w/v* reducing sugars. The highest alcohol content (11.8% *v/v*) and the lowest sugar content (0.4% *w/v*) were recorded with 10%

*w/v* inoculum after 20 days of fermentation. The same experimental variant obtained the highest marks in the sensory analysis for appearance, colour, taste and mouthfeel.

The overall quality of fermented alcoholic beverages from figs is influenced by the type of raw materials and the form in which they are used. A study performed by Jeong and their team in 2005 [10] compared the effects of using fresh sliced figs, fig juice or frozen figs, fermented by *S. cerevisiae* yeast, with and without the addition of 100 ppm dried fig leaf powder. The drink obtained from fresh fruits had an alcohol concentration of 12.1% *v/v*, pH 3.91, total acidity 0.42% *w/v* and a rest of 9.9 °Brix. Sensory analysis showed that drinks made from cut fruits were superior to those made from fruit juice, while those made from frozen figs were rated similarly to those made from fresh fruits. The addition of leaf powder improved the quality of all drinks [10]. The variety of figs used also has major influence on the quality of beverage. For distillates obtained from fig fruits, both the variety and the form in which figs are used are determinant for the volatile profiles. Clearly distinct distillates were obtained from monovarietal fresh figs, as compared to distillates from pluri-varietal dehydrated figs. In fig distillates, 130 volatile compounds were identified, of which 18 were common regardless of the form of raw material, fresh or dried, esters being the most represented [28].

Lactic fermentation also has positive effects on fig drinks. Lactic fermentation contributes to the increase of the total content of polyphenols (up to five times as compared to unfermented fig juice) and preserves or increases the antioxidant capacity of fig juice [9], depending also on the bacterium used (*Lactobacillus bulgaricus*, *L. plantarum* or *L. acidophilus*). As well, lactic fermentation of fig juice, using *Lactobacillus casei*, *Lactobacillus plantarum* and *Lactobacillus delbrueckii*, lead to the production of a fermented beverage with probiotic qualities. Lactic bacteria developed well in fig juice, all species studied reaching 8–9 log CFU mL$^{-1}$ after 48 h at 30 °C, decreasing after 4 weeks of storage at a low temperature of 4 °C to 6 and 5 log CFU mL$^{-1}$ for *L. delbrueckii* and *L. plantarum*, respectively. *L. casei* decreased more rapidly, reaching 3 log CFU mL$^{-1}$ after only 2 weeks. According to the sensory analysis (general appearance, taste, smell, consistency), the beverage obtained with *L. casei* was the most appreciated [8].

Acetic fermentation was also performed to obtain some home-made vinegars based on different recipes starting from either fresh figs, dehydrated and fresh figs, fresh figs and apple cider vinegar or fresh figs and grapes. Depending on the raw material and the fermentation time, a mix of microorganisms were found to determine the fermentation: acetic bacteria 2.68–8.23 log CFU mL$^{-1}$, lactic acid bacteria 0.81–8.20 log CFU mL$^{-1}$, mesophilic aerobic bacteria 2.26–7.29 log CFU mL$^{-1}$ and yeasts 0.00–6.49 log CFU mL$^{-1}$. The chemical characteristics of the vinegars vary as follows: pH 3.05–3.73; total acidity 0.210–0.697 g L$^{-1}$; volatile acidity 0.197–0.646 g L$^{-1}$; ash 1.11–5.60 g L$^{-1}$; density 1.0002–1.1448 and alcoholic strength <0.5% *v/v* [7].

As presented, there are only a few scientific studies dealing with the possibility of obtaining fermented products from figs (studies in references [7–10,25,27]), and the production of wine-like beverages is even less investigated (studies in references [10,25,27]).

Therefore, the present study on the production of fermented alcoholic beverages from three varieties of figs aims to achieve a technological process for the production of a quality fermented alcoholic beverage. The chemical composition and sensory properties of the products are also evaluated.

## 2. Materials and Methods

### 2.1. Fig as the Raw Material for the Preparation of an Alcoholic Fermented Beverage

The figs were harvested in an orchard from Şviniţa Commune, Mehedinţi County, located in Southwest Romania. There, the figs usually ripen in the middle to end of August. The figs used in the experiment were harvested on 21 August 2020, transported in plastic boxes to Bucharest where they were processed on 25 August 2020. Being a perishable fruit, the degradation process started during the transportation period, but before processing the healthiest fruits were selected and used.

The fig genotypes are as follows:

1.  Variety G1, Șvinița Black—purple epicarp and pink flesh (Figure 1a);
2.  Variety G2, Șvinița Yellow—yellow epicarp and yellow pulp (Figure 1b);
3.  Variety G3, Șvinița Green—green-brown epicarp and amber pulp (Figure 1c).

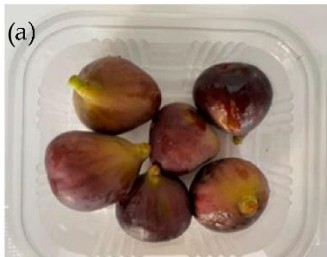 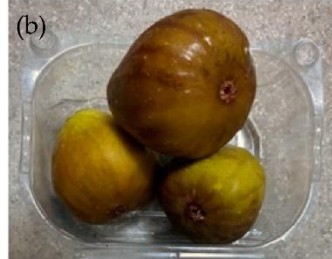 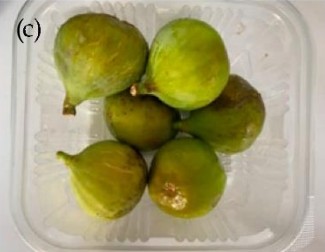

**Figure 1.** Fig varieties: (**a**) Șvinița Black, (**b**) Șvinița Yellow and (**c**) Șvinița Green.

For the raw fruits physicochemical parameters were determined in accordance with the descriptions in Section 2.3.1 and results reported in Table 1.

**Table 1.** Main physicochemical parameters for the figs used to produce fermented alcoholic beverages (values ± standard errors).

| Fig Physicochemical Parameters * | Fig Variety G1 | Fig Variety G2 | Fig Variety G3 |
|---|---|---|---|
| Skin colour | purple | yellow | green |
| Mass (g/fruit) | 20.44 ± 0.95 [a] | **47.49 ± 2.25 [b]** | 33.39 ± 2.70 [c] |
| TSS (°Brix) | 13.95 ± 0.59 [a] | 15.07 ± 0.39 [a] | 13.98 ± 0.30 [a] |
| Acidity (malic acid g kg$^{-1}$) | **2.22 ± 0.04 [a]** | 1.19 ± 0.02 [b] | 1.05 ± 0.03 [c] |
| pH | 4.40 ± 0.02 [a] | 4.95 ± 0.04 [b] | **5.13 ± 0.03 [c]** |

\* Different letters on each row show that there is a significant difference between the averages for the assessed varieties at a probability level of 95% ($\alpha = 0.05$) determined by one-way ANOVA and Tukey test. The averages with the highest values, if significantly different from those in other samples, are marked in bold. TSS represents total soluble substance.

### 2.2. Fig Processing for the Production of the Fermented Alcoholic Beverage

The beverage was prepared for each variety in triplicate, by batch fermentation, following a protocol similar to those used for winemaking [26]. However, because the crushed figs are not juicy enough, water was also added to obtain a fermentable fluid. For our beverage commercially available Zizin still mineral water (Romania) was used, having a chemical composition in minerals, as average values in mg L$^{-1}$ as follows: Ca (73.65), Mg (3101), Na (8655), pH (7.2), HCO$_2$ (250.1), SO$_2$ < 40, chlorides (16.38), NO$_2$ < 0.05, K (1.08), fixed residue at 180 °C (233). As well, to obtain a beverage with a higher alcohol content, which is necessary to be able to preserve it for longer times, the natural sugar of figs was also supplemented with sugar beet saccharose. The use of yeasts, enzymes, sulphites, the correction of acidity are oenological practices which were also used in our beverage production protocol.

For each batch 2 kg of figs were crushed and mixed with 6 L of a sugar solution of 33.33% *w/v* concentration (2 kg of sugar, 4 L of water), thus obtaining 8 L of dense homogeneous mash. Taking into account that the initial sugar concentration from figs, measured as the total soluble solids content, was about 14–15 Brix, depending on genotype, the final sugar concentration in the mash before fermentation reached approximately 29 °Brix, allowing for a potential alcohol content of 17% *v/v*, if fermented completely.

To ensure a good balance of sugar–acidity, the total acidity of the fig mash was corrected for each batch by adding 1 g L$^{-1}$ tartaric acid (Enartis, San Martino, Italy). For antioxidant protection the unfermented mash was sulphited with 20 mL of 10% *w/v* solution of potassium metabisulphite (K$_2$S$_2$O$_5$ from Enartis, San Martino, Italy), which

releases in acidic media about 50% $w/v$ of sulphur dioxide. For a better extraction and release of the varietal aromatic components from the fig skins, as well as for clarification of the mash, 0.04 g L$^{-1}$ of pectolytic enzyme Zimared Plus (Enologica Vason, San Pietro in Cariano, Italy), containing pectolytic cellulase, hemicellulase, protease, polygalacturonase, and pectin lyase was added to each sample.

To obtain beverages with a good aromatic profile, the start and control of the fermentation was performed in the presence of a *Saccharomyces cerevisiae* wine yeast Excellence B2 Elegant White (Lamothe-Abiet, Canéjan, France), which was purchased as a dry yeast and used after rehydration in warm water. Also, at the beginning of fermentation a nutrient and bioregulator V Activ Premium (Enologica Vason, San Pietro in Cariano, Italy), containing 40% $w/w$ yeast autolysates, 29.94% $w/w$ cellulose, 30% $w/w$ diammonium hydrogen phosphate and 0.06% $w/w$ thiamine was also added with the yeast, to ensure the completion of fermentation, especially taking into account the high amount of sugar required to be fermented. The dose of dry yeast used was 0.2 g L$^{-1}$ and that of the nutrient was 0.4 g L$^{-1}$.

The alcoholic fermentation was carried out in 10 L containers at a temperature of 24 $\pm$ 1 °C.

Each sample was shaken three times a day for homogenisation and to enhance extraction. After 27 days, when the fermentation stopped and the dead yeast settled down, the beverages were racked to separate the limpid fluid from the lees and transfer it in small bottles of 2.0 L. For antioxidant protection and microbiological stabilization, 1.5 mL L$^{-1}$ of a 10% $w/v$ solution of potassium metabisulphite was added to each container in which the racked beverage was transferred. The bottles were filled completely, to avoid contact with air and prevent oxidation.

### 2.3. Analyses

#### 2.3.1. Determination of Chemical Characteristics of Fig Fruits

To determine the main parameters of the fresh figs, 15 fruits were randomly selected from each variety, their average mass measured and then crushed using a Retsch GM 200 grinding mill (Retsch, Haan, Germany) until a paste was obtained. This paste was then used for total soluble substance (TSS), acidity and pH determination.

For the determination of TSS, a digital refractometer KRUSS OPTRONIC DR301-95 (Hamburg, Germany) was used. Three aliquots of each variety of fig paste were measured by refractometer and the results expressed as the average of the three measurements. The instrument lens was washed with distilled water and wiped before each reading. The determined values, in °Brix units, are generally related to the sugar content.

For total acidity determination a titrimetric method was applied. Three fig paste aliquots were taken from each fig variety, weighed using a Kern analytical balance (Balingen, Germany), and submitted to titration with a 0.1 M solution of NaOH, using an automatic titrator TitroLine easy (Karl Fischer, Pforzheim, Germany). The acidity was expressed in g of malic acid kg$^{-1}$. The results are calculated according to the following formula:

Titratable acidity (g kg$^{-1}$ malic acid) = (V × N × C × 1000) m$^{-1}$

V—NaOH volume used for titration, N—normality of NaOH solution, C—the gram equivalent of malic acid (0.067), m—sample mass

The pH was measured potentiometrically using a Seven Excellence multiparameter device (Mettler Toledo, Ohio, USA). Before each measurement, the electrode was washed with double-distilled water and dried with a paper towel.

The pH and °Brix values were measured in the juice prepared for fermentation and during the entire process of fermentation.

#### 2.3.2. Determination of the Chemical Characteristics of the Fermented Alcoholic Beverages from Figs

To determine the main parameters of the fermented alcoholic beverage from figs, the usual methods applied for wine (fermented alcoholic beverage from grapes) were used. The pH was determined by direct measurement as described in Section 2.3.1; the alcoholic concentration was determined using the Simple Distillation Method (STAS 6182/6-70);

volatile acidity by using the Titrimetric Method with phenolphthalein as indicator (STAS 6182/2-86); reducing sugars were determined using the SCHOLL Iodometric Method (STAS 6182/18-8); the free sulphur dioxide and total sulphur dioxide were measured using the Direct Dosage Iodometric Method (STAS 6182/13-72). These methods are Romanian standard methods for the wine industry from the Romanian National Standardization Body-ASRO and are fully harmonized with the European Commission Regulations (EC) No 1293/2005 [29] and (EC) No 606/2009 [30].

### 2.3.3. Sensory Analyses of Fermented Alcoholic Beverages from Figs

The sensory analyses were performed by a group of 22 evaluators (11 women and 11 men) who were informed that three types of fermented alcoholic beverages from figs are presented, but no other details were specified, the samples being served anonymized.

To obtain quantitative data for the evaluation, a specially designed score sheet was used, based on a modified tasting sheet registered as part of a patent in a previous project [31]. The score sheets were on paper and the scales on which the tasters marked the level of intensity of the perceived characteristics (acidity, sweetness, bitterness, mouthfeel, colour intensity, aroma intensity) are 100 mm long, so the marks were easily transformed into values between 0 and 100, by direct measurement with a ruler. For several aroma classes, on the same score sheet discontinuous scales were provided, with values between 1 and 5 (1—least intense, 5—most intense), which were later multiplied by 20 to calibrate also up to 100 points. Moreover, each type of identified aroma was freely described by each taster on their score sheet. All data obtained were recorded in tables and subsequently analysed and interpreted.

### 2.3.4. Statistical Analyses

The data were statistically analysed with the software package Origin 9.0 from Origin-Lab (Northampton, MA, USA) and Excel from Microsoft Office Professional Plus 2019 (Microsoft Corporation, Redmond, WA, USA). All the replicates analysed were averaged and the significance of the mean differences were calculated by one-way ANOVA or by the Kruskal–Wallis non-parametric test (one-way ANOVA on ranks), when the data did not have a normal distribution, as determined by the Shapiro–Wilks Normality Test. To compare the means for all the three varieties, Tukey or Duncan post-hoc analyses were performed, and the values that were significantly different at $p < 0.05$ were indicated with different letters. Principal component analysis (PCA) computed with Origin 9.0 was applied for the sensory analysis data, to reduce the large number of data and determine the main aroma descriptors which correlate to the aromatic profile of each variety.

## 3. Results

### 3.1. Chemical Parameters of the Figs Used to Produce Fermented Alcoholic Beverages

The physicochemical parameters for the raw materials are presented in Table 1, assessed for 15 fig fruits from each variety.

The differences in these varieties' main characteristics are determinant for the quality of the alcoholic beverages produced by fermenting them.

### 3.2. Fermentation Evolution and Parameters of the Alcoholic Beverages Resulting from Figs

### 3.2.1. Evolution of the Sugar Content during the Fermentation to Produce Alcoholic Beverages from Figs

The easiest way to measure the sugar content of the fig mash submitted to fermentation is the determination of the total soluble solids content expressed in °Brix. In the fruit mash, especially after the water, sugar and tartaric acid were added before fermentation (as described in Section 2.2), the main content of the soluble solids is glucose and fructose, which are used by the fermentation yeasts and converted to ethanol and other secondary products.

After the inoculation with the fermentation yeast, the refractometric measurement was performed regularly, twice a day in the first 13 days, and only once a day in the following 14 days, when the fermentation rate became very slow.

In the fermentation diagram shown in Figure 2, it can be observed that all the varieties had a similar evolution of the sugar conversion into ethanol and $CO_2$. The TSS, measured daily for each variety and each repetition, constantly decreased with the progression of fermentation, but no statistically significant differences were apparent among varieties, by applying the ANOVA test at $p < 0.05$.

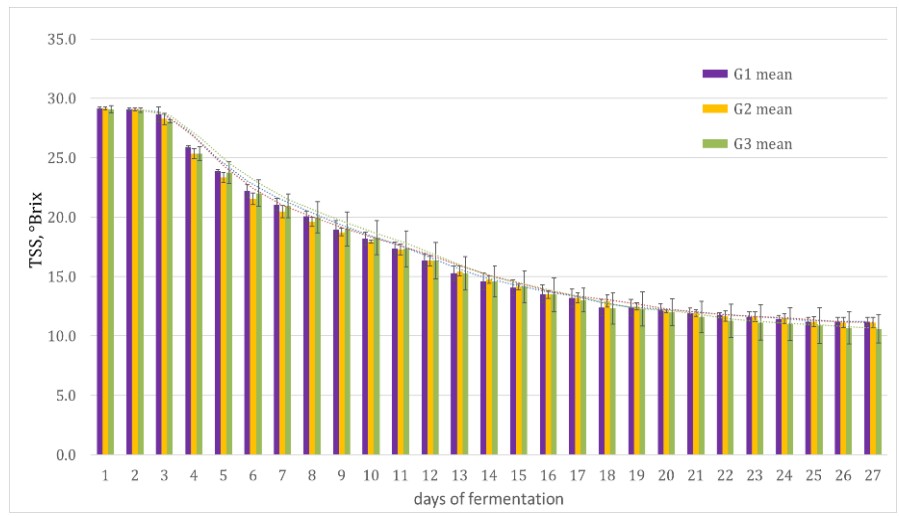

**Figure 2.** Evolution of the total soluble solids during fermentation of each fig variety (TSS average of 3 repetitions ± standard deviation).

### 3.2.2. Chemical Parameters of the Alcoholic Beverages Resulting from Figs

After stabilization and racking from the solid parts, the fermented alcoholic beverages were analysed to determine their main physicochemical parameters. The results for each variety's beverage are included in Table 2.

**Table 2.** Main physicochemical parameters of the fermented alcoholic beverages from the three varieties of figs.

| Fig Beverage Physicochemical Parameter | Fig Variety G1 | Fig Variety G2 | Fig Variety G3 |
|---|---|---|---|
| alcohol degree (% *v/v*) | **18.00 ± 0.09** [a] | 17.36 ± 0.09 [b] | 17.48 ± 0.08 [b] |
| reducing sugar (g L$^{-1}$ glucose + fructose) | **19.90 ± 0.91** [a] | 26.98 ± 1.80 [b] | 12.59 ± 0.43 [c] |
| total acidity (g L$^{-1}$ tartaric acid and meq L$^{-1}$) | 4.87 ± 0.11 [a] | 5.06 ± 0.06 [a] | 5.02 ± 0.08 [a] |
| | 64.93 ± 1.47 [a] | 67.46 ± 0.80 [a] | 66.93 ± 1.07 [a] |
| malic acid (g L$^{-1}$ and meq L$^{-1}$) | 0.34 ± 0.05 [a] | 0.42 ± 0.02 [a] | 0.43 ± 0.01 [a] |
| | 5.08 ± 0.75 [a] | 6.27 ± 0.30 [a] | 6.42 ± 0.15 [a] |
| lactic acid (g L$^{-1}$ and meq L$^{-1}$) | 0.23 ± 0.03 [a] | 0.21 ± 0.03 [a] | 0.17 ± 0.02 [a] |
| | 2.56 ± 0.33 [a] | 2.33 ± 0.33 [a] | 1.89 ± 0.22 [a] |
| volatile acidity (g L$^{-1}$ acetic acid and meq L$^{-1}$) | 0.55 ± 0.05 [a] | 0.49 ± 0.04 [a] | 0.53 ± 0.02 [a] |
| | 9.17 ± 0.83 [a] | 8.17 ± 0.67 [a] | 8.83 ± 0.33 [a] |
| pH | 3.41 ± 0.01 [a] | 3.37 ± 0.01 [ab] | 3.35 ± 0.01 [b] |
| free $SO_2$ (mg L$^{-1}$) | 4.07 ± 0.98 [a] | 4.94 ± 1.00 [a] | 3.27 ± 0.65 [a] |
| total $SO_2$ (mg L$^{-1}$) | 37.43 ± 1.14 [a] | 37.20 ± 3.70 [a] | 32.73 ± 3.72 [a] |
| K$^+$ mg L$^{-1}$ | 177.25 ± 18.74 [a] | 164.50 ± 5.07 [a] | 167.75 ± 16.00 [a] |
| Ca$^{2+}$ mg L$^{-1}$ | 51.58 ± 4.15 [a] | 57.10 ± 4.95 [a] | 53.25 ± 2.52 [a] |
| Density | **0.9929 ± 1.25 × 10$^{-4}$** [a] | 0.9949 ± 6.58 × 10$^{-4}$ [b] | 0.9912 ± 8.74 × 10$^{-4}$ [b] |

\* Different letters on each row show that there is a significant difference between the averages of the same parameter for beverages resulting from different varieties, determined by one-way ANOVA and the Tukey test at a probability level of 95% (α = 0.05). The averages with the highest values, if significantly different from those in other samples, are marked in bold.

3.2.3. Sensory Features of Fermented Alcoholic Beverages from Figs

The characteristics of the fermented alcoholic beverages from figs were evaluated as described in Section 2.3.3. and the average values of the perceived intensity for the main sensory parameters are presented in Table 3.

**Table 3.** General parameters assessed by sensory analysis for the fig fermented beverages on scales of 100 points (average $\pm$ standard errors).

| Fig Beverage Sensory Parameters | Fig Variety G1 | Fig Variety G2 | Fig Variety G3 |
|---|---|---|---|
| Acidity | 45.91 $\pm$ 4.49 [a] | 50.68 $\pm$ 3.95 [a] | 47.27 $\pm$ 3.72 [a] |
| Sweetness | 50.45 $\pm$ 4.36 [a] | 46.59 $\pm$ 4.31 [a] | 46.59 $\pm$ 3.97 [a] |
| Bitterness | 40.68 $\pm$ 4.28 [a] | 40.91 $\pm$ 3.77 [a] | 37.05 $\pm$ 2.86 [a] |
| Mouthfeel (extract) | **58.18 $\pm$ 2.36 [a]** | 49.32 $\pm$ 3.21 [ab] | 46.14 $\pm$ 2.42 [b] |
| Colour intensity | **67.27 $\pm$ 2.67 [a]** | 39.32 $\pm$ 3.30 [b] | 41.59 $\pm$ 3.46 [b] |
| Aroma intensity | 63.64 $\pm$ 4.06 [a] | 59.32 $\pm$ 3.87 [a] | 52.95 $\pm$ 3.69 [a] |

\* Different letters on each row show that there is a significant difference between the averages for the assessed varieties at a probability level of 95% ($\alpha = 0.05$) determined by one-way ANOVA and the Tukey test. The averages with the highest values, if significantly different from those in other samples, are marked in bold.

The statistical tests, one-way ANOVA as well as the Kruskal–Wallis test ($p < 0.05$) showed that the variety induced differences in the perception of the colour intensity—as well as the mouthfeel, which is correlated with the total extract of the sample.

These main sensory characteristics profile of all beverages produced from our varieties are included in Figure 3, for an easier comparison.

Interestingly, women and men perceived the beverages characteristics and their balance differently, in accordance with their experiences and preferences. Thus, in Figure 4 the profiles of the main sensory characteristics of the three types of beverages, as determined by the women and men tasters, are compared. The green lines represent profiles provided by the women's group and the red lines profiles from the men's group. The beverages from various varieties are differentiated by the type of line used (dotted for G1, dashed for G2 and continuous for G3).

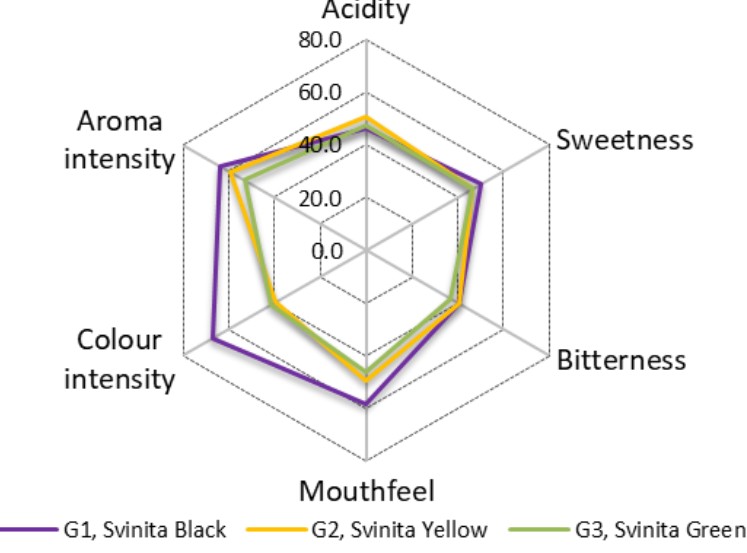

**Figure 3.** The average sensory profiles containing the main sensory characteristics of the alcoholic beverages fermented from three different fig varieties.

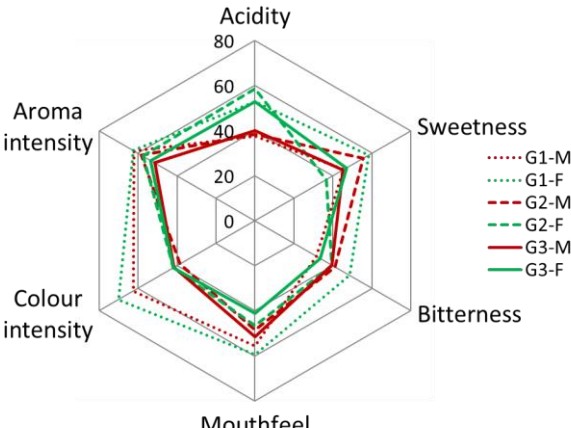

**Figure 4.** Gender comparisons of sensory perception of beverages from G1 (dotted lines), G2 (dashed lines) and G3 (continuous line); green lines—women's perception (F), red lines—men's perception (M).

For each alcoholic beverage fermented from figs the tasters identified several aroma attributes for which they also provided an evaluation of the intensity of perception.

Several attributes were identified by the tasters to describe the fig alcoholic beverages. Of the total 14 descriptors used, 8 of them (apple, grapefruit, peach, quince, raisins, pear, honey and lemon) were found in all three products. Fig and bitter almond aroma were identified in the green and yellow varieties, grass only in the green variety and strawberry and raspberry only in the black variety. Wherever possible, the aroma descriptors were classified either as varietal aroma or fermentation aroma (Figures 5 and 6). Aromas of figs, pears, raisins and honey were included in the "varietal aroma" group, based on their similarity with the sensory perception from tasting the fresh fig fruits. The "fermentation aroma" group included aromas of apples, quince and peach, generally caused by ester compounds formed during fermentation [26] and grapefruit notes caused by thiol compounds resulted from enzymic breakdown of precursors during fermentation [26].

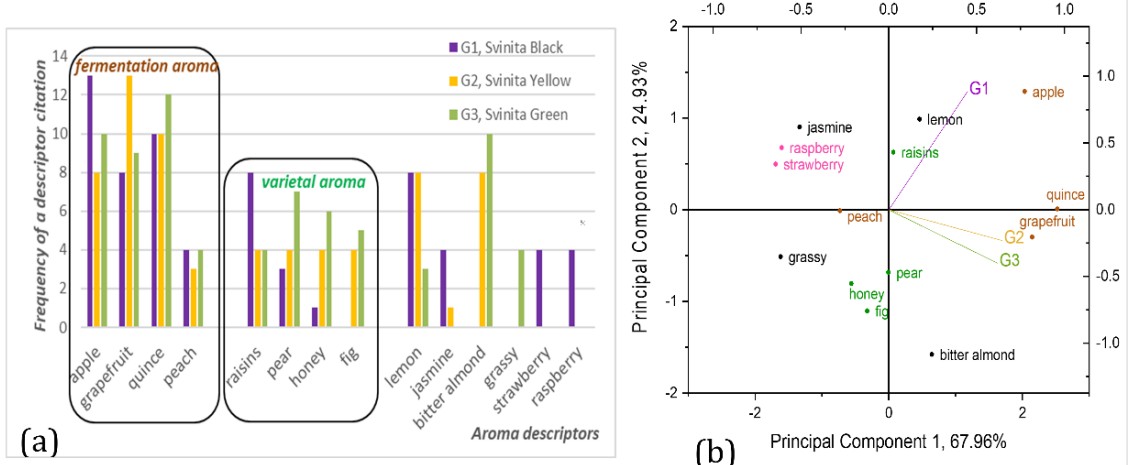

**Figure 5.** Frequency of aroma descriptor citations (**a**) and principal component analysis based on the frequency of aroma descriptor citations (**b**) for the beverages from fig varieties G1–G3.

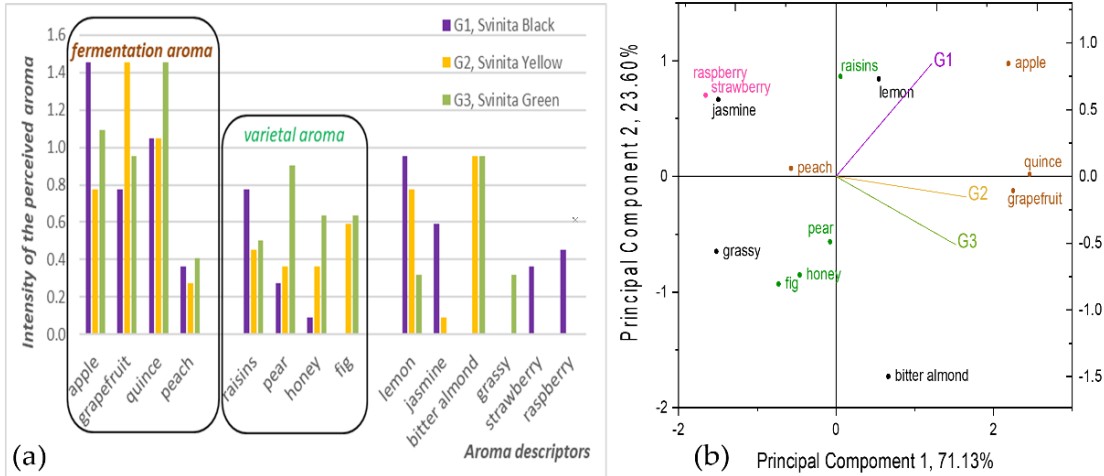

**Figure 6.** Intensity of the perceived aroma attribute (**a**) and principal component analysis based on the perceived intensity of aroma attribute (**b**) for the beverages from fig varieties G1–G3.

In accordance with the frequency of each descriptor cited by the tasters, in Figure 5a all 14 aroma attributes are presented for all the three fig varieties. The principal component analysis in Figure 5b presents the correlation of the aroma descriptors with the overall aroma perception of each beverage obtained from the three fig varieties, G1–G3.

The fine balance between these aroma attributes also has a big influence on the overall flavour, therefore, the intensity with which each aroma is perceived is determinant. The principal component analysis in Figure 6 shows the intensities perceived for the main aroma attributes (Figure 6a) and the correlation of these aroma types with the fig variety from which the alcoholic beverage is obtained (Figure 6b).

Based on the perception intensity of the main varietal and fermentation aroma attributes, the aroma profiles of the obtained beverages are illustrated in Figure 7a.

To have an overall description of the alcoholic beverages obtained from the three fig varieties, the total sensory profiles containing both the main sensory parameters and the aroma parameters are illustrated in Figure 7b.

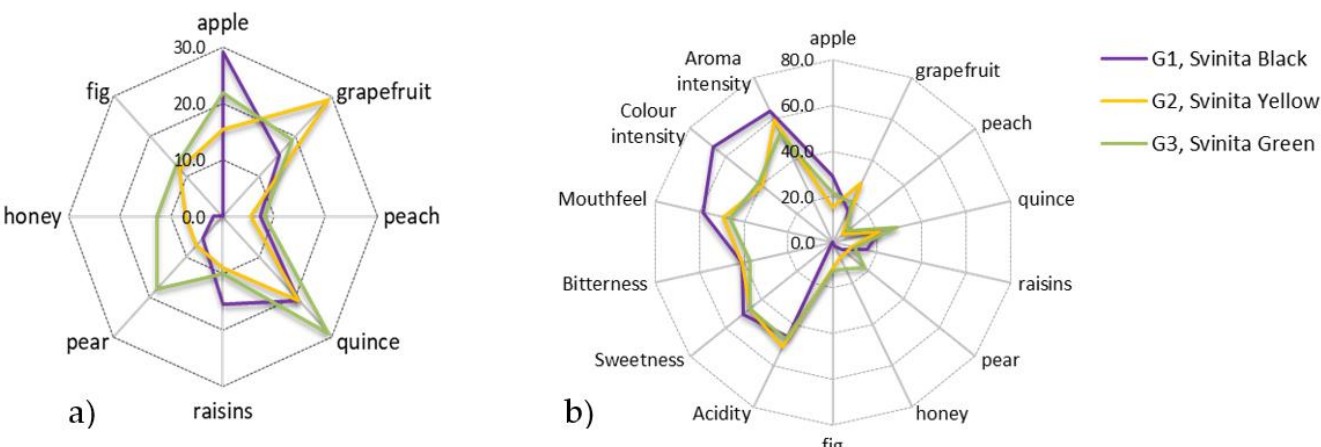

**Figure 7.** Aroma profiles (**a**) and total sensory profiles (**b**) of the alcoholic beverage produced from the three fig varieties.

## 4. Discussion

### 4.1. Chemical Parameters of the Figs Used to Produce Fermented Alcoholic Beverages

The fruit mass and the colour of skins are factors which influence the extraction of colour pigments and other compounds found in the skins, a bigger skin surface/volume

ratio being in favour of more concentration of skin compounds in the fruit mash. With the smallest fruit (favourable skin surface/volume ratio) and purple skin colour (more colour pigments), the variety Şviniţa Black is expected to lead to the most coloured fermented beverage. Furthermore, as seen in Table 1, it is also the variety with the highest acidity, and this helps to preserve the beverage's colour vivacity and taste freshness. The variety Şviniţa Yellow has the biggest fruit, which in turn leads to more diluted skin extracted compounds in the beverage and a lighter colour. The variety Şviniţa Green has medium-sized fruit and lowest acidity (highest pH), which would also correlate with lighter colour and taste.

However, in order to obtain better quality fermented beverages some of the original parameters of these fruits will be corrected, by adjusting the acidity and sugar levels. The sugar concentration is not significantly different in the three fruit varieties, and through complete fermentation it will lead to an alcohol concentration of only about 8.1% *v*/*v* on average (ranging from 7.8 to 8.4% *v*/*v* in accordance with the rules of transformation of °Brix into potential alcohol, AWRI, 2016, [32]), which is relatively low for a stable beverage meant to be preserved without many other technological interventions. Thus, the simpler way is to correct the sugar and acidity, similarly with the wine technology, by adding sugar and some tartaric acid, as described in Section 2.2.

### 4.2. Fermentation Evolution and Parameters of the Alcoholic Beverages Resulting from Figs

4.2.1. Evolution of the Sugar Content during the Fermentation to Produce Alcoholic Beverages from Figs

The comparable behaviour observed in Figure 2 regarding the fermentation progress was to be expected, considering that the fig pastes used to obtain the alcoholic beverages had similar initial sugar values, irrespective of the variety. However, it should be noticed that the yeast was not able to entirely ferment the carbohydrates present in figs, the fermentation slowing down very much in the last 10 days and remaining constant in the days 26–27 to an average of TSS of about 11 °Brix, this relatively high value being likely determined by the presence of the non-fermentable polysaccharides and yeast residues.

This also shows that under the same conditions the technology of preparation of the fermented alcoholic beverage from figs is reproducible and does not depend much on the fig variety.

4.2.2. Chemical Parameters of the Alcoholic Beverages Resulting from Figs

As regards the main physicochemical parameters of the fermented alcoholic beverages included in Table 2, with the exception of the alcohol, remaining sugar and density, the main parameters are not significantly different in beverages obtained from the three fig varieties. Total acidity ranges between 64.9–67.5 meq $L^{-1}$ (4.9–5.1 g $L^{-1}$ expressed as tartaric acid), of which 5.1–6.4 meq $L^{-1}$ is malic acid, 1.9–2.6 meq $L^{-1}$ is lactic acid and 8.2–9.2 meq $L^{-1}$ is acetic acid. The main part of the acidity is represented by tartaric acid coming from the acidity correction performed before fermentation, which gives a very good sensory balance with the remaining sugars, which are at a level of a semi-sweet wine. Because the tartaric acid is preponderant in the total acidity of the fig beverage, in Table 2 total acidity is reported in units of tartaric acid. The extraction of potassium and calcium was similar irrespective of the fig variety, ranging between 165–177 mg $L^{-1}$ for $K^+$ and 52–57 mg $L^{-1}$ for $Ca^{2+}$. The sulphur dioxide is at low limits, a bit above 30 mg $L^{-1}$ for total $SO_2$ and less than mg $L^{-1}$ for the free $SO_2$, which shows that most of the used $SO_2$ reacted and remained in the solid parts of the fig mash. Normally, these levels of sulphur dioxide would not ensure sufficient antioxidant and antimicrobial protection on a wine, but for the present fig beverages, given their much higher alcohol content (17.5–18.0 versus 11.0–13.0% *v*/*v* in most wines), higher doses of $SO_2$ would not be necessary. The presence of only low levels of $SO_2$ is important for the consumers, who tend to avoid additives in food and beverages.

Considering that the level of sugars was also corrected before fermentation, the alcohol in the obtained beverages reached the level of 17.5–18.0 *v*/*v*. After that, the yeast could not

tolerate the alcohol concentration and stopped working, leaving some unfermented sugars. The sugar remaining in each beverage is correlated with the sugar content of each fig variety, as the correction of sugar was similarly done for all varieties and the fermentation performed with the same yeast. Being over 12 g L$^{-1}$ and under 45 g L$^{-1}$, the sugar values of all fig beverages are in the range of a semi-sweet grape wine (EC 1308/2013) [33].

4.2.3. Sensory Features of Fermented Alcoholic Beverages from Figs

Regarding the general parameters assessed by sensory analysis and included in Table 3, as expected, the pink salmon colour of the beverage resulting from G1 (Șvinița Black) was perceived as being more intense than the straw yellow of G2 (Șvinița Yellow) or amber colour of G3 (Șvinița Green). G3 colour intensity was perceived as being higher than in the case of G2, but the difference did not reach statistical significance. The mouthfeel of G1 was clearly perceived as being higher than the one of G3, with G2 placing itself in between the other varieties, not being significantly different than either of them. This perception for G1, both regarding the colour and the mouthfeel, is determined by the presence of more polyphenols extracted from the figs, such as anthocyanins (colour pigments) and procyanidins (tannins).

As it can also be observed in Figure 3, irrespective of the fig variety used, the resulting beverages were well balanced in taste, with acidity, sweetness and bitterness not differing significantly from one another and being placed around the middle of the 100-point scale in the sensory profile web graph, meaning that the taste is mild and palatable.

The sensory perception of women and men was slightly different, very likely being influenced by the personal preferences and experiences. From the sensory profiles drawn in Figure 4, it can be noticed that, compared to men, irrespective of the beverage tasted, women generally perceived the acidity as being higher and the sweetness and mouthfeel lower. This confirms that women generally consider that a beverage is more balanced if the sugar content is higher, which in turn attenuates the perception of acidity. The observation is important and should be kept in mind when the products are made for a certain market segment.

Regarding the aroma profile, there is a tendency for the variety G1 to transmit a higher aroma intensity than the other two varieties, but this was not confirmed statistically (Table 3). Even though the overall aroma intensity of the samples was not perceived as being different among the beverages from different varieties, the aroma and flavour quality and profile showed major differences. These aroma differences were evaluated based on the frequency of aroma descriptor citations by the tasters (Figure 5), as well as based on the intensity of each perceived aroma attribute (Figure 6).

In accordance with the sensory analysis, the most cited aroma attributes are those induced by the fermentation. They are also among the most intensely perceived by the tasters, showing that fermentation can majorly influence the quality of the beverage produced. In Figure 5a,b these aroma attributes generated as a result of fermentation are the apple, grapefruit, quince and peach, mostly in this order or intensity.

Some varietal aroma is also preserved in the final beverages, tasters identifying aromas described as raisins, pear honey and fig fruit (Figure 5a,b). Raisin fragrance and flavour is mostly correlated with the black coloured variety G1, while the aroma of fresh fig is identified only in the case of the less coloured varieties, G2 and G3.

Both PCA analyses (Figures 5b and 6b) are very useful to determine the influence of the variety and the fermentation of the final aroma of the beverage.

The distribution of aroma attributed in the PCA diagrams show that the Principal Component 1, which represents more than 67% of the total variance (67.96% in the case of the average of perceived aroma intensity and 71.13% in the case of the frequency with which the aroma descriptor was mentioned) incorporates mainly aroma induced in the beverage by the alcoholic fermentation-apple, grapefruit, quince, peach. Therefore, it can be considered that PC1 separates the beverages in accordance with the fermentation aroma. These fermentation aroma attributes are presented in the PCA diagrams with a brown font.

Conversely, the principal component 2, which represents about 24% of the total variance (24.93% in the case of the perceived intensity of aroma and 23.60% in the case of the frequency of the descriptor citation) is rather correlated with the aroma reminiscent of the fig fruit. Thus, it can be considered that PC2 separates the beverages in accordance with the varietal aroma attributes-raisins, pear, honey, fresh fig. These fermentation aroma descriptors are presented in the PCA diagrams with green font.

The other aroma attributes, found in the PCA charts with black font, are either correlated with both the fruit and the fermentation, or are specific to certain fig varieties. The lemon attribute, which is usually mentioned due to the acidity perceived rather than the specific aroma of the lemon peel, is influenced by the fruit acidity, by the correction of acidity before fermentation and by the fermentation itself. The aroma of raspberry and strawberry are mostly perceived in the case of the G1 beverage, which is derived from the variety G1 with purple skin colour, these red-fruit aromas being usually associated with pink or red-coloured beverages. Bitter almond, grassy notes and jasmine flower notes have some singular influences depending on the fig variety: grass in G3, bitter almond in G2 and G3, jasmine in G1 and G2.

Apart from the positive aroma, some low-intensity off-flavours were also perceived by some tasters. G1 reminded tasters of dry hay due to the raw tannins extracted during the long fermentation (27 days), some yeast off-favour and hydrogen sulphide were perceived in G2 due to the long contact with the yeast during the long fermentation and a burning sensation and some ethyl acetate were noticed in G3, which are usually correlated with fermentation at warm temperatures. Thus, for a better quality of the beverages the fermentation time and temperature should be reduced.

## 5. Conclusions

As a result of this research, it was proven that good quality fermented alcoholic beverages can be obtained from figs. By correcting the sugar and acidity initial parameters of the raw material, the fermentation process is proceeding smoothly and reliably, irrespective of the fig variety used.

All the resulting beverages were well-balanced, with mild main characteristics, such as sweetness, acidity, bitterness and mouthfeel.

The variety had a major influence on the aroma and overall quality of the final beverage, but fermentation is even more important, the production of a specific aroma during fermentation induces the main differences in the aroma profiles of the beverages.

Among the varieties studied, G1 was the most appreciated, especially due to its pleasant salmon-pink colour, but also due to a higher aromatic intensity. However, the other two varieties have also some specific pleasant aroma attributes, being thus able to diversify the offer of this kind of beverage.

The technology of these beverages can still be improved, by reducing to the minimum the required external interventions for composition corrections. As compared to this preliminary experiment the initial quantity of sugar in the recipe should be decreased, in order to lower the alcohol in the final beverage by 2–3% $v/v$, which in turn will also reduce the long fermentation period and the contact with yeasts. Applying selected yeasts and lowering the fermentation temperature will further improve the products. It should also be kept in mind that the fruits to be turned into alcoholic beverages should be fresh and not touched by decay.

As demonstrated, the farmers who cultivate figs can valorise their production by also preparing quality fermented alcoholic beverages from their figs, to diversify their offer and also to reduce the loss caused by the depreciation of the fruits which cannot be sold immediately on the fresh fruit market.

**Author Contributions:** Conceptualization, A.O.A.; methodology, A.O.A. and E.M.; software and formal analysis, A.O.A.; investigation, E.M. and A.O.A.; resources, E.M. and A.O.A.; data curation, A.O.A.; writing—original draft preparation, E.M. and A.O.A.; writing—review and editing, A.O.A.; supervision, A.O.A. All authors have read and agreed to the published version of the manuscript.

**Funding:** This research received no external funding.

**Data Availability Statement:** Not applicable.

**Acknowledgments:** The research was performed with the administrative and technical support (equipment and raw materials) belonging to the University of Agronomic Sciences and Veterinary Medicine of Bucharest, Romania. The results were communicated as Eposter Flash Presentation at the International Horticultural Congress 2022 (IHC 2022) Angers, France.

**Conflicts of Interest:** The authors declare no conflict of interest.

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
