# Peer review of "Figs (Ficus carica L.) Used as Raw Material for Obtaining Alcoholic Fermented Beverages"

_beverages, doi:10.3390/beverages8040060_

Round 1
Reviewer 1 Report
The manuscript is interesting. Some corrections should be done to improve the quality.
Pg 5 – line 224 – In formula “Titratable acidity g kg-1 malic acid) =” please correct it to: “Titratable acidity (g kg-1 malic acid) =”
Pg. 6, inside Table 1, please correct TSS (Brixº) to TSS (Brix)
Pg. 8, inside Table 2, please replace “alcohol” by “alcohol degree”. Authors should explain why total acidity of Fig beverages was expressed as tartaric acid.
Pg. 9, Inside Figure 3, Please correct the word “Colour intenity”. In Figure 4, “Color intensity” should be “Colour intensity” or in both figures, change to “Color”. Please verify also the text.
Pg. 10, Figure 5 and 6. Authors should explain the classification of fermentation and varietal aroma for those sensory descriptors.
Pg. 11, Figure 7. In the b) profile should be avoided the aroma descriptors. And please correct “Colour intenity”.
Pg. 14, lines 590-591: In Conclusions, there is a sentence: “Applying selected yeasts and lowering the fermentation temperature will further improve the products.” In materials and methods section, authors should insert the fermentation temperature of the experiments.
Reviewer 2 Report
The authors should address the following comments:
Line 15. The sentence is incomplete.
Line 49. What do the authors mean by the phrase ‘unimportant amount’?
Line 51. What are these other components?
Line 52-54. Rephrase sentence for clarity. Fragmentation of the sentence may be helpful
Line 93. Delete the word ‘made’
Line 97. The alcohol content of 11.9% ABV is out of the range quoted in line 96
Line 106. Give pH value to 1 decimal and throughout the article
Line 137. The assertion of ‘few studies’ is not well qualified. How many studies do the authors consider to be few?
Line 166-167. Cite the protocol referred to in the text
Line 182. How is complete fermentation ascertained?
Line 205. Delete the article ‘the’ after the word avoid
Line 218. Punctuate properly
Line 266. MS Excel not properly cited as Origin is
Line 281 (Table 1). Include TSS in the table legend and give the pH to one decimal
The authors corrected the pH and sugar before fermentation. Was the effect of the correction on the sensorial attributes of the beverage investigated? If so, what were the findings?
Reviewer 3 Report
The paper entitled: "Figs (Ficus carica L.) used as raw material for obtaining alcoholic fermented beverages ", describes experiments on the production of alcoholic beverages fermented from fresh figs and assessment their quality and acceptability. Generally, the authors presented the research results in an appropriate manner using well-chosen methods. However, here are few points which must be considered during revision of the manuscript:
Line 141 It is not true that authors have carried out optimization of process of figs-based mashes fermentation. This sentence needs improvement.
Line 188 It is worth adding the enzymatic activities of the enzyme preparation used in the mashing.
Lines 193-195 Add the applied doses of yeast and nutrient.
Line 220 Convert concentration units from N to M (molarity)
Line 248 More information about evaluators is required (how many women/men, age)
Line 271 What software was used to do PCA analysis?
Subsection 3.2.2. Lack description of the results from Table 2.
Subsection 3.2.3. Lack description of the results from Table 3.
Discussion needs to be improved. More literature references are recommended.
Note to the whole text: complete the concentrations expressed as % with w/w, w/v, v/v, respectively.
Round 2
Reviewer 3 Report
The manuscript has been improved according the reviewers' recommendations, and it may be accepted for publication in Beverages journal.